# Modeling the spectrum and determinants of multimorbidity risk among older adults in India

Ajay Kumar[1]*, Bharti Singh[2]

**1** Department of Biostatistics and Epidemiology, International Institute for Population Sciences (IIPS), Mumbai, Maharashtra, India, **2** Department of Survey Research and Data Analytics, International Institute for Population Sciences (IIPS), Mumbai, Maharashtra, India

* rpvvajaykumar@gmail.com

## Abstract

### Background

India is passing through a parallel phase of demographic and epidemiological transition coupled with the shifting burden of multimorbidity. Unhealthy ageing and escalating morbidity burden have been identified as key drivers of this shifting multimorbidity risk among older adults in India. This study aims to assess the distribution of morbidities and multimorbidity, provide new estimates of multimorbidity risk by socioeconomic and demographic factors and further evaluate the multimorbidity count risk conditioned on leading factors.

### Methods

This study used the nationally representative Longitudinal Ageing Study in India (LASI), Wave – 1, 2017–18, data of individuals aged 45 years and above. First, we assessed the relative proportional share of morbidities and compositions of multimorbidity counts over age. Second, we applied the Random Forest (RF) model to estimate the age-specific risk of multimorbidity susceptibility associated with socio-economic and demographic factors over age. Finally, conditional plots were constructed to assess the distributional composition of the leading factors affecting multimorbidity counts.

### Results

The prevalence of multimorbidity was 43.20%. Eye disorders, followed by cardiovascular disease (CVDs), had the highest proportional share over age. Endocrine diseases, Gastrointestinal Conditions, and Infectious diseases showed a concordant decreasing proportional share in later age. The relative share of five or more multimorbidity counts increased significantly with age. The median expected risk of multimorbidity was significantly higher in females (66 years) than in males (71 years). The study also provides empirical evidence that individuals with higher levels of

**Data availability statement:** The datasets used for this study are publicly available and can be accessed through: https://www.iipsdata.ac.in/datacatalog_detail/5LASI-data by filling out a data request form along with a valid identity proof

**Funding:** The author(s) received no specific funding for this work.

**Competing interests:** The authors have declared that no competing interests exist.

education, obesity, currently working, and poor childhood health were more prone to higher risk of multimorbidity at an early age. Furthermore, obesity was significantly associated with early multimorbidity onset and led to a pronounced escalation of complex multimorbidity progression, particularly in females.

## Conclusions

Collective public health interventions are crucial to address early multimorbidity onset and burden disparities, to promote healthier ageing, and to address etiological factors.

---

## 1 Introduction

Omran's theory outlines India's health transition from the "Age of Receding Pandemics" to the "Age of Degenerative and Man-Made Diseases," highlighting a shift from communicable diseases (CDs) to non-communicable diseases (NCDs) [1,2]. This epidemiological transition is accompanied by rapidly changing lifestyles, urbanisation, and better healthcare facilities, which has led to reduced case fatality and mortality rates over decades [3–5]. Nonetheless, the proportion of life spent with NCDs and disability substantially increased, which resulted in the 'Morbidity Expansion' and characterized by wide span of prolonged morbidities and a huge disease burden in India [6,7].

In recent decades, morbidity expansion patterns have been further compounded by elevated multimorbidity (coexistence of two or more conditions/diseases in an individual) burden in Low and Middle-Income countries (LMIC) [8–11]. However, multimorbidity gained global attention for its adverse effects on economic, social, and public health domains [9,10,12]. It strains healthcare systems, increases levels of polypharmacy, costs, and resources while amplifying health inequality and reducing the productivity of individuals and society as a whole by prolonging the burden of disease [10,13]. Moreover, the critical implications of multimorbidity include declining functional limitations, reduced quality of life, elevated mortality risks, and behavioural risk factors with psychological antecedents and consequences, placing additional responsibility on individuals and their caregivers [14–16]. Thus, the output leads to poor economic status due to the individual's reduced working years and the additional burden on caregivers [17,18].

Understanding the scale and distribution of multimorbidity is crucial, as its prevalence varies widely across regions and populations. LMIC showed a trend of higher burden due to the cumulative impact of socio-economic and demographic risk factors and limited healthcare resources coupled with the rapid transition from infectious to NCDs [8,19]. Currently, India is undergoing thorough the parallel phase of epidemiological and demographical transition, with multimorbidity emerging as a significant concern that contributes to the accelerated progression of unhealthy ageing [20]. The recent Longitudinal Ageing Study of India (LASI) reported that nearly one-third of the population aged 45 and older are suffer from one of the five common chronic

morbidities (hypertension, diabetes, cardiovascular diseases, chronic respiratory diseases, arthritis) [21], while among 15–44 years, the burden of multimorbidity was seven percent [22].

The key determinants of multimorbidity are socio-economic, demographic, behavioural, and environmental factors [9,23]. In the Indian context, the rise in unhealthy ageing has become a leading factor of multimorbidity [24]. As age increases, physiological changes and accumulated exposure throughout life increase susceptibility to chronic diseases [25]. Further, occupational hazards have emerged as key determinants for the Indian population. According to the Food and Agriculture Organization (FAO) report, seventy percent of India's rural households still depend primarily on agriculture for their livelihood, with eighty-two percent of farmers being small and marginal [26,27]. Such dependence exposes individuals to extensive physical labour, increasing the likelihood of workplace accidents that can result in lifelong disabilities. According to the National Multidimensional Poverty Index, approximately fifteen percent of India's population is in a state of multidimensional Poverty [28,29]. Limited financial resources constrain access to healthcare services, medications, early detection, timely treatment, and preventive measures [13,15,18,24]. Moreover, inadequate living conditions and sanitation increase the risk of disease transmission and exacerbate existing health conditions [30,31]. The vicious cycle of poverty and poor health perpetuates as individuals struggle to break free from the burden of multimorbidity and socio-economic deprivation. For the affluence class, it has been found that lifestyle factors such as an unhealthy diet, physical inactivity, and substance use have been the reasons for the early onset of chronic conditions [32,33].

The disease burden in India is substantially shaped by its societal framework, with social dynamics interacting with environmental and behavioural elements [2]. While previous studies have often examined social factors in isolation, recognising their interrelation is crucial, particularly in the context of multimorbidity [22]. Despite an observed increase in morbidity levels over the past two decades, there remains a significant gap in understanding the true changes in multimorbidity risk and distributional composition. As the prevalence of multimorbidity rises, it is imperative to adapt the healthcare system to address these complexities [34]. This highlights the urgent need for evidence-based policies and programs tailored to effectively address multimorbidity challenges, especially for the older population. Hence, the objective of our study is to estimate the pattern, risk, and leading factors of multimorbidity in India, given the understanding of the multimorbidity burden among the ageing population using the Random Forest (RF) model [35]. Unlike traditional statistical methods, RF can accommodate high-dimensional data and non-linear relationships, without imposing prior assumptions. This makes it particularly well-suited for analysing large scale complex healthcare datasets, where interactions between multiple predictors and risk outcomes are dynamic, given the multifaceted nature of multimorbidity.

## 2 Materials and methods

### 2.1 Data

We used the Longitudinal Ageing Study in India (LASI, Wave 1, 2017–18) dataset, which provides comprehensive details on an extensive number of morbidities for individuals 45 years and older and their spouses irrespective of age and their socio-economic and demographic details, covering 28 states and 8 Union Territories of India [36]. LASI is a longitudinal survey that adopted a multi-stage stratified areas probability cluster sampling design, with three stages in rural areas and four stages in urban areas. The present study utilised merged information from the individual and biomarker datasets. The dataset contained 73,396 individuals, and this study focused on the information of 59,830 individuals who were 45 years and older.

**2.1.1 Data availability and ethical consideration.** Prior informed consent (written and verbal) from all participants was obtained by field survey agencies. LASI administered consent forms at the household and individual levels following the Human Subject Protection. The Indian Council of Medical Research (ICMR) extended the necessary guidance and mandatory ethical approval to conduct the LASI survey. The Institutional Review Board at International Institute for Population Sciences (IIPS) provided additional approval for the study protocol. All methods were carried out under relevant guidelines and regulations of the ICMR.

## 2.2 Variables definitions

**2.2.1 Morbidities.** The study encompassed a total of twelve major morbidities and chronic conditions using the information on all self-reported and diagnosis available observations in LASI, which were documented based on responses to the question, "Has any health professional ever diagnosed you with the following chronic conditions or diseases?" and "In the past 2 years, have you had any of the following diseases?". These morbidities or chronic condition are Musculoskeletal disorders (Arthritis, Rheumatism, Osteoporosis), CVDs or cardiovascular disease (Hypertension, Heart attack, Heart blockage, Heart failure, Arrhythmias, Heart rheumatic, Heart congenital or Structural disorders, Stroke), Chronic lung diseases (Asthma, Bronchitis, COPD (Chronic Obstructive Pulmonary Disease)), Eye disorders (Cataract, Glaucoma, Hypermetropia, Myopia, Presbyopia), Neurological or Psychiatric disorder (Depression, Dementia, Neurological disorders, Psychiatric disorders), Endocrine diseases (Diabetes), Infections (Jaundice or Hepatitis, Tuberculosis) and other specific morbidities and disorder are Cancer, High cholesterol, Hearing disorders, Gastrointestinal conditions, and Urogenital diseases.

**2.2.2 Outcome variable.** We calculated the prevalence and proportional share of twelve morbidities and chronic conditions. Initially, all twelve morbidities and chronic conditions were categorized as binary variables (absent or present). Subsequently, the total number of morbidities was aggregated and similarly classified into a binary form:

1] No Multimorbidity (individuals who did not have combinations of two or more morbidities or chronic conditions)

2] Multimorbidity (individuals who had combinations of two or more morbidities or chronic conditions).

**2.2.3 Exogenous factors.** We considered childhood health as one of the variables in addition to socio-economic and demographic variables. We have adopted Anderson's healthcare utilisation framework for the selection and categorising of the independent variables in two groups (35).

1) Predisposing factors: Age, Sex, Caste group, Religion, and Marital status

2) Enabling factors: Highest level of schooling, Monthly per capita consumption expenditure (MPCE) quintiles, Residence, Regions, Working status, Childhood health, Self-rated health, Physical activity, Tobacco consumption, Alcohol consumption and Body Mass Index (BMI).

Further, detailed descriptions of all the exogenous factors are available in the S2 Table.

## 2.3 Statistical analysis

First, we computed the weighted prevalence of morbidities and multimorbidity to assess the individual morbidity and multimorbidity burden among older adults in India. Second, we assessed the relative share of morbidities and multimorbidity over age using stacked area and bar plots.

The approach of prior studies in India for multimorbidity risk estimations was built on odds ratio or relative risk models, which were not suitable for understanding the non-linear relationship between multimorbidity and background characteristics. Hence, we applied the non-parametric Random Forest (RF) method to estimate the risk of multimorbidity associated with socio-economic and demographic risk factors over age, with no prior covariate distribution assumptions [37].

A random forest of 300 decision trees was constructed with D bootstrap samples, and the Gini index was used as the node-splitting criterion [38], and used to measure the heterogeneity of a dataset at a node. It helps determine the best feature and threshold for splitting a node by minimizing impurity, thereby creating more homogeneous child nodes which is given by

$$Gini(D) = 1 - \sum_{i=1}^{c} pi^2$$

Where $p_i$ is the proportion of the class in dataset "D" and "c" is the number of classes.

Further probability (risk) estimation of multimorbidity was calculated for each age over 45–95 years. Such that in the RF model of M decision trees, for each tree $T_i$, let the prediction be $\hat{y}_i$, where $\hat{y}_i$ is the predictor class of $T_i$.

Hence, to estimate the risk of class $C_k$ for at a given instance x is given by:

$$\hat{P}(C_k|x) = \frac{1}{M}\sum_{i=1}^{M} 1(\hat{y}_i - C_k)$$

Where,

The Indicator function is 1 if the prediction of tree $T_i$ in class $C_k$ and 0 otherwise. M is the total number of trees in the RF. $\hat{P}(C_k|x)$ is the estimated probability (risk) that instance x belongs to the class $C_k$.

Thus, the ensemble risk was determined by averaging the tree estimators of the risk output. Moreover, the mean and median expected risk by age were estimated for all exogenous factors. The Mann-Whitney-U test was used at a 95% level of significance for the association within exogenous factors. Finally, conditional plots were used across all ages to understand the interdependence distribution among leading socio-economic and demographic factors. All analyses were performed using SAS version 9.4.M8 (SAS Institute; Cary, NC), R Studio version 2023.12.1＋402 (www.R-project.org) and STATA 17.

## 3 Results

### 3.1 Relative share and patterns of morbidities and multimorbidity

Fig 1 shows that across all age groups, eye disorders had the highest relative share, followed closely by CVDs. The proportion of endocrine diseases, gastrointestinal conditions, and infectious diseases showed a decreasing share in later age, suggesting that these may be more common in middle age or early older adulthood. In contrast, hearing disorders showed the diverging morbidity proportions across all the age groups. Notably, more severe diseases, such as Cancer and Chronic lung diseases, accounted for the lowest share of morbidities across all age groups, suggesting either lower prevalence, underdiagnosis, or higher mortality associated with these diseases.

S1 Table and Fig 1 further illustrates the prevalence of both morbidities and multimorbidity, with 43.2% (red bubble) of individuals experiencing multimorbidity in the older age. Eye disorders (48.01%) were the most prevalent morbidity, followed by cardiovascular diseases (28.31%), gastrointestinal conditions (18.29%), and musculoskeletal disorders (16.33%). Endocrine diseases (11.88%) and chronic lung diseases (6.61%) were also significant, while severe diseases and conditions like cancer (0.62%) and neurological or psychiatric disorders (2.31%) were less common.

Fig 2 depicts the proportional share of multimorbidity counts over 45 years and above age groups. The proportion of any two to four multimorbidity counts was relatively higher than in higher-order complex multimorbidity configurations. However, their share almost remained constant across all age groups. In contrast, the relative proportion of any five or more multimorbidity counts increased significantly with age, reaching its peak in the 70–74 age group. Notably, the highest contribution of higher-order multimorbidity counts was observed in the 70–74 age group. Moreover, S3 Table. presents the prevalence of multimorbidity among older adults in India, stratified by exogenous factors or socio-economic and demographic factors.

### 3.2 Risk of multimorbidity by exogenous risk factors

Table 1 illustrates the estimates of the age-group specific, mean and median expected risk of multimorbidity susceptibility across socio-economic and demographic risk factors, while Fig 3 illustrates the age-specific line plot of expected risk over exogenous factors.

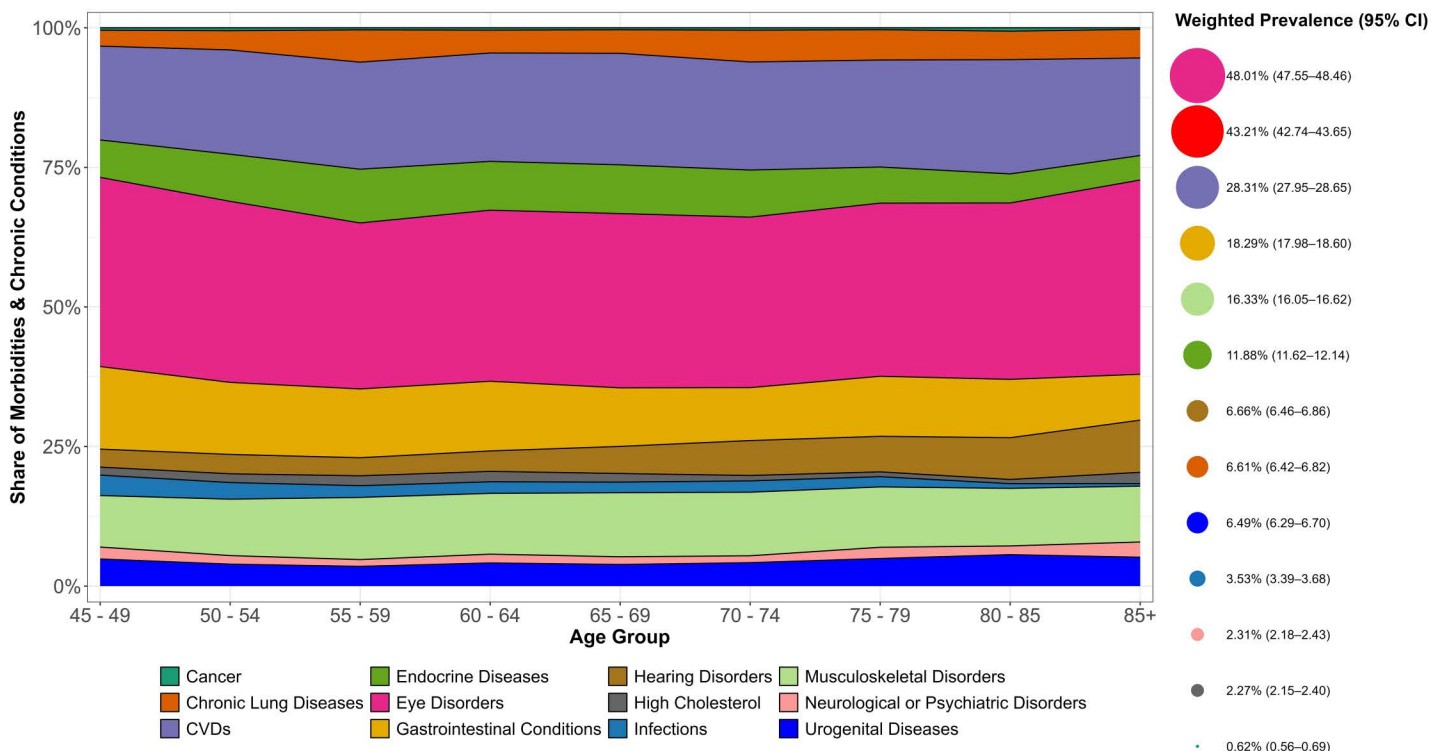

**Fig 1. Relative proportional distribution and prevalence rates of morbidities and chronic conditions among older adults by age groups, LASI Wave-1, 2017-18, India.**

Women had a higher risk of multimorbidity over age compared to their counterparts. The mean and median risks of multimorbidity were 0.52 and 0.59 in women versus 0.46 and 0.49 in men. The risk gradient was steeper in women compared to men. Women exhibited multimorbidity expected risk of 0.25, 0.50, and 0.75, at ages 55, 66, and 75 years, respectively. Meanwhile, men showed the multimorbidity expected risk of 0.21, 0.44 and 0.69 at the same ages, indicating men of the same age as women had a lower risk of multimorbidity. In the urban area, the multimorbidity risk was higher (mean (0.53) and median (0.59)) compared to rural areas (mean (0.48) and median (0.52)). Residents residing in urban areas showed a stronger differential in oldest of old ages compared to older-adult ages (Fig 3.b).

Among the predisposing factors, marital status (Fig 3.h) exhibited substantial differentials in multimorbidity expected risk. Currently married and divorced/separated persons showed the multimorbidity mean risk of 0.53 and 0.51, respectively, which was higher than the mean risk of multimorbidity in widowed persons (0.47). By religion (Fig 3.e), Hindus (0.49) and Christians (0.49) had slightly lower multimorbidity mean risks than Muslims (0.49) and others. The regional (Fig 3.i) differentials in multimorbidity were narrow. The western region showed the highest mean risk of multimorbidity (0.54) and the central region showed the lowest mean risk of multimorbidity (0.47). In addition, caste categories (Fig 3.f) also showed narrow differentials in multimorbidity. Schedule Caste population had a higher mean risk of multimorbidity (0.52) compared to Other Backward Classes (0.49), Scheduled Tribes (0.48) and other castes (0.50).

The MPCE quintiles (Fig 3.c) also did not show strong differentials in the risk of multimorbidity. Sorted by MPCE, the mean risk of multimorbidity was highest at 0.52 in the richest quintile compared to 0.47 in the poorest quintile. The gradient of risk of multimorbidity remains similar across economic classes. Further, the age variation in multimorbidity by economic class was also similar. The gaps in the risk of multimorbidity between these income quintiles remained unchanged

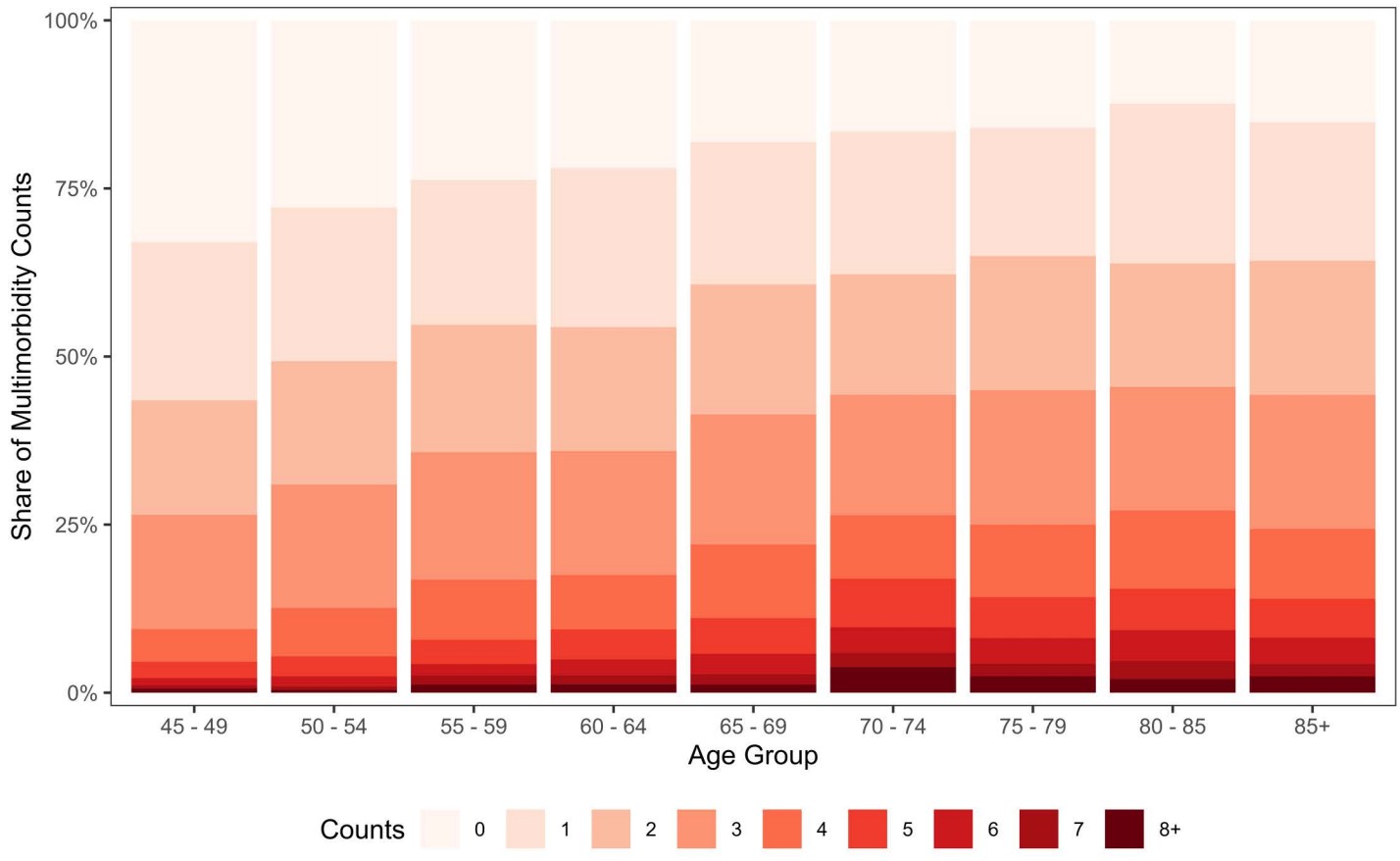

**Fig 2. Relative proportional distribution of multimorbidity counts among older adults by age groups, LASI Wave-1, 2017–18, India.**

up to the oldest of old ages. Nonetheless, the risk of multimorbidity was more than 0.50 for all MPCE classes in the age group of 70–74 years and older. On the other hand, the categories of working status (Fig 3.g) exhibited larger differentials in the risk of multimorbidity. Currently, working persons (0.52) showed a higher risk of multimorbidity compared to those who have never worked (0.49) and are currently not working (0.45). The differentials by categories of working status were stronger in older-adult age groups versus in old age groups. Years of schooling (Fig 3.d) revealed stronger differentials in the risk of multimorbidity. The mean risk of multimorbidity was 0.57 in persons with 10 or more years of schooling compared to no schooling (0.46). For persons with 10 or more years of schooling, the risk of multimorbidity was 0.50 at 65 years. Whereas, for persons with no schooling, the risk was 0.50 at 70 years. There was a difference of five years for the same level of risk in multimorbidity between 10 or more years of schooling and no schooling. However, the differentials in multimorbidity were maintained among oldest of old adults.

Childhood health (Fig 3.l) status revealed stronger differentials in multimorbidity risk. The mean risk of multimorbidity was 0.53 for very poor childhood health and 0.49 for very good childhood health. Physical activity (Fig 3.n) showed expected differentials in multimorbidity. Persons with casual physical activity (0.53) showed a higher risk of multimorbidity, whereas those engaging in physical activity every day or weekly (0.49) showed a lower risk of multimorbidity. In the oldest of the old age group, physical activity was compromised. Therefore, the gradient of risk of multimorbidity was similar between physically active and inactive persons. Nonetheless, regular physical activity provided an advantage in lowering the risk of multimorbidity.

**Table 1. Age-group specific, mean, and median expected risk of multimorbidity susceptibility across socio-economic and demographic sub-groups among adults aged 45 years and older, LASI Wave-1, 2017–18, India.**

| Exogenous factors | Expected Risk | | | | | | | | | | | |
|---|---|---|---|---|---|---|---|---|---|---|---|---|
| | 45–49 | 50–54 | 55–59 | 60–64 | 65–69 | 70–74 | 75–79 | 80–84 | 85 + | Mean | Median | *p-value* |
| **Sex** | | | | | | | | | | | | |
| Male | 0.03 | 0.10 | 0.17 | 0.27 | 0.40 | 0.53 | 0.65 | 0.75 | 0.83 | 0.46 | 0.49 | <0.001 |
| Female | 0.06 | 0.15 | 0.25 | 0.37 | 0.50 | 0.62 | 0.72 | 0.80 | 0.85 | 0.52 | 0.59 | |
| **Residence** | | | | | | | | | | | | |
| Rural | 0.04 | 0.12 | 0.20 | 0.30 | 0.44 | 0.56 | 0.67 | 0.76 | 0.83 | 0.48 | 0.52 | <0.001 |
| Urban | 0.05 | 0.14 | 0.23 | 0.35 | 0.50 | 0.63 | 0.73 | 0.81 | 0.87 | 0.53 | 0.59 | |
| **MPCE Quintile [a]** | | | | | | | | | | | | |
| Poorest | 0.04 | 0.11 | 0.19 | 0.29 | 0.42 | 0.55 | 0.66 | 0.76 | 0.82 | 0.47 | 0.51 | <0.001 |
| Poorer | 0.04 | 0.12 | 0.20 | 0.31 | 0.44 | 0.57 | 0.68 | 0.77 | 0.84 | 0.49 | 0.53 | |
| Middle | 0.04 | 0.12 | 0.21 | 0.32 | 0.45 | 0.58 | 0.69 | 0.78 | 0.84 | 0.49 | 0.54 | |
| Richer | 0.05 | 0.13 | 0.22 | 0.33 | 0.47 | 0.59 | 0.70 | 0.79 | 0.85 | 0.50 | 0.55 | |
| Richest | 0.05 | 0.14 | 0.23 | 0.35 | 0.49 | 0.61 | 0.72 | 0.80 | 0.86 | 0.52 | 0.58 | |
| **Highest level of Schooling** | | | | | | | | | | | | |
| No Schooling | 0.03 | 0.10 | 0.17 | 0.27 | 0.40 | 0.53 | 0.64 | 0.74 | 0.82 | 0.46 | 0.49 | <0.001 |
| <5 Years | 0.04 | 0.12 | 0.21 | 0.32 | 0.46 | 0.58 | 0.69 | 0.78 | 0.85 | 0.50 | 0.54 | |
| 5–9 Years | 0.05 | 0.15 | 0.25 | 0.37 | 0.51 | 0.64 | 0.74 | 0.82 | 0.87 | 0.53 | 0.60 | |
| 10 + Years | 0.07 | 0.18 | 0.29 | 0.41 | 0.56 | 0.69 | 0.78 | 0.85 | 0.89 | 0.57 | 0.65 | |
| **Religion** | | | | | | | | | | | | |
| Hindu | 0.04 | 0.12 | 0.20 | 0.31 | 0.44 | 0.57 | 0.68 | 0.77 | 0.84 | 0.49 | 0.53 | <0.001 |
| Muslim | 0.06 | 0.15 | 0.25 | 0.37 | 0.51 | 0.64 | 0.74 | 0.81 | 0.86 | 0.53 | 0.60 | |
| Christian | 0.04 | 0.12 | 0.21 | 0.32 | 0.45 | 0.57 | 0.68 | 0.76 | 0.83 | 0.49 | 0.53 | |
| Other | 0.05 | 0.14 | 0.24 | 0.36 | 0.49 | 0.62 | 0.72 | 0.81 | 0.86 | 0.52 | 0.57 | |
| **Caste Category [b]** | | | | | | | | | | | | |
| ST | 0.04 | 0.12 | 0.20 | 0.31 | 0.44 | 0.56 | 0.67 | 0.76 | 0.83 | 0.48 | 0.52 | <0.001 |
| SC | 0.05 | 0.14 | 0.24 | 0.36 | 0.50 | 0.62 | 0.72 | 0.80 | 0.85 | 0.52 | 0.59 | |
| OBC | 0.04 | 0.12 | 0.21 | 0.32 | 0.45 | 0.58 | 0.69 | 0.78 | 0.84 | 0.49 | 0.54 | |
| Other | 0.04 | 0.12 | 0.21 | 0.32 | 0.45 | 0.58 | 0.69 | 0.78 | 0.85 | 0.50 | 0.54 | |
| **Working Status** | | | | | | | | | | | | |
| Never worked | 0.04 | 0.12 | 0.20 | 0.31 | 0.45 | 0.58 | 0.69 | 0.78 | 0.85 | 0.49 | 0.54 | <0.001 |
| Currently working | 0.05 | 0.15 | 0.26 | 0.37 | 0.51 | 0.63 | 0.73 | 0.80 | 0.85 | 0.52 | 0.59 | |
| Currently not working | 0.03 | 0.08 | 0.14 | 0.23 | 0.37 | 0.52 | 0.64 | 0.75 | 0.84 | 0.45 | 0.47 | |
| **Current Marital Status** | | | | | | | | | | | | |
| Currently married | 0.05 | 0.14 | 0.23 | 0.35 | 0.49 | 0.61 | 0.71 | 0.80 | 0.85 | 0.51 | 0.57 | <0.001 |
| Widowed | 0.03 | 0.10 | 0.17 | 0.28 | 0.41 | 0.54 | 0.66 | 0.76 | 0.83 | 0.47 | 0.50 | |
| Divorced/separated/other | 0.06 | 0.16 | 0.26 | 0.38 | 0.52 | 0.64 | 0.74 | 0.81 | 0.86 | 0.53 | 0.60 | |
| **Region** | | | | | | | | | | | | |
| North | 0.04 | 0.12 | 0.20 | 0.31 | 0.45 | 0.57 | 0.68 | 0.78 | 0.84 | 0.49 | 0.53 | <0.001 |
| Central | 0.04 | 0.11 | 0.19 | 0.29 | 0.42 | 0.54 | 0.65 | 0.74 | 0.81 | 0.47 | 0.50 | |
| East | 0.05 | 0.13 | 0.22 | 0.33 | 0.46 | 0.59 | 0.69 | 0.78 | 0.84 | 0.50 | 0.55 | |
| Northeast | 0.04 | 0.12 | 0.21 | 0.31 | 0.44 | 0.56 | 0.67 | 0.76 | 0.83 | 0.48 | 0.52 | |
| West | 0.05 | 0.14 | 0.24 | 0.36 | 0.51 | 0.64 | 0.75 | 0.83 | 0.88 | 0.54 | 0.60 | |
| South | 0.05 | 0.13 | 0.22 | 0.34 | 0.48 | 0.61 | 0.72 | 0.81 | 0.87 | 0.52 | 0.57 | |

*(Continued)*

**Table 1.** (Continued)

| Exogenous factors | Expected Risk | | | | | | | | | | | |
|---|---|---|---|---|---|---|---|---|---|---|---|---|
| | 45–49 | 50–54 | 55–59 | 60–64 | 65–69 | 70–74 | 75–79 | 80–84 | 85 + | Mean | Median | *p-value* |
| **Alcohol Consumption** | | | | | | | | | | | | |
| Lifetime abstainer | 0.04 | 0.12 | 0.20 | 0.31 | 0.45 | 0.57 | 0.68 | 0.77 | 0.84 | 0.49 | 0.53 | 0.01 |
| Infrequent non-heavy drinker | 0.05 | 0.14 | 0.24 | 0.35 | 0.49 | 0.61 | 0.71 | 0.80 | 0.85 | 0.52 | 0.57 | |
| Frequent non-heavy drinker | 0.05 | 0.15 | 0.24 | 0.36 | 0.50 | 0.63 | 0.73 | 0.80 | 0.86 | 0.52 | 0.59 | |
| Heavy episodic drinker | 0.05 | 0.14 | 0.24 | 0.35 | 0.49 | 0.61 | 0.71 | 0.79 | 0.85 | 0.51 | 0.57 | |
| **Tobacco Consumption** [c] | | | | | | | | | | | | |
| Lifetime abstainer | 0.04 | 0.12 | 0.21 | 0.31 | 0.45 | 0.57 | 0.68 | 0.77 | 0.84 | 0.49 | 0.53 | 0.177 |
| Smokes tobacco | 0.04 | 0.12 | 0.21 | 0.33 | 0.46 | 0.59 | 0.70 | 0.78 | 0.84 | 0.50 | 0.55 | |
| Smokeless tobacco | 0.05 | 0.13 | 0.22 | 0.33 | 0.46 | 0.59 | 0.69 | 0.78 | 0.84 | 0.50 | 0.55 | |
| Both | 0.05 | 0.13 | 0.22 | 0.34 | 0.47 | 0.60 | 0.70 | 0.79 | 0.85 | 0.51 | 0.56 | |
| **Childhood Health** | | | | | | | | | | | | |
| Very good | 0.04 | 0.12 | 0.20 | 0.31 | 0.44 | 0.57 | 0.68 | 0.77 | 0.84 | 0.49 | 0.53 | <0.001 |
| Good | 0.04 | 0.12 | 0.21 | 0.32 | 0.45 | 0.58 | 0.69 | 0.78 | 0.84 | 0.49 | 0.54 | |
| Fair | 0.05 | 0.14 | 0.23 | 0.35 | 0.49 | 0.61 | 0.71 | 0.80 | 0.85 | 0.51 | 0.57 | |
| Poor | 0.06 | 0.17 | 0.27 | 0.39 | 0.53 | 0.65 | 0.74 | 0.82 | 0.87 | 0.54 | 0.61 | |
| Very Poor | 0.06 | 0.16 | 0.26 | 0.37 | 0.51 | 0.63 | 0.73 | 0.81 | 0.86 | 0.53 | 0.59 | |
| **BMI** [d] | | | | | | | | | | | | |
| Underweight | 0.04 | 0.10 | 0.17 | 0.27 | 0.40 | 0.52 | 0.64 | 0.74 | 0.81 | 0.46 | 0.48 | <0.001 |
| Normal | 0.04 | 0.11 | 0.19 | 0.29 | 0.42 | 0.55 | 0.66 | 0.76 | 0.83 | 0.47 | 0.51 | |
| Overweight | 0.05 | 0.15 | 0.26 | 0.39 | 0.54 | 0.67 | 0.77 | 0.84 | 0.89 | 0.55 | 0.64 | |
| Obese | 0.07 | 0.19 | 0.31 | 0.45 | 0.60 | 0.72 | 0.81 | 0.87 | 0.91 | 0.59 | 0.68 | |
| **Physical Activity** | | | | | | | | | | | | |
| Everyday | 0.05 | 0.13 | 0.22 | 0.34 | 0.48 | 0.61 | 0.71 | 0.79 | 0.85 | 0.51 | 0.56 | <0.001 |
| Weekly | 0.04 | 0.11 | 0.20 | 0.30 | 0.44 | 0.57 | 0.68 | 0.77 | 0.84 | 0.49 | 0.53 | |
| Casual | 0.06 | 0.15 | 0.25 | 0.37 | 0.51 | 0.64 | 0.74 | 0.81 | 0.86 | 0.53 | 0.60 | |

[a]- Wealth index as Monthly per capita consumption expenditure (MPCE),

[b]- Scheduled Castes (SC), Scheduled Tribes (ST), Other Backward Classes (OBC)

[c]- Current user;

[d]- Body mass index as Underweight (BMI ≤ 18.4 kg/m²), Normal (18.5 kg/m2 ≤ BMI ≤ 24.9 kg/m²), Overweight (25.0 kg/m² ≤ BMI ≤ 29.9 kg/m²), Obese (BMI ≥ 30 kg/m²).

In comparison to predisposing and enabling factors, the risk factors such as BMI (Fig 3.m) compared to alcohol (Fig 3.j) and tobacco consumption (Fig 3.k) showed greater differentials in the risk of multimorbidity. The mean risk of multimorbidity was 0.51 in tobacco users, including smokeless tobacco and cigarettes etc. and heavy alcohol consumers. Whereas, the mean risk of multimorbidity was 0.55 and 0.59 in overweight and obese persons, respectively. When it comes to the factors that increase the risk of developing multimorbidity individual's BMI is more evident than the smoking habits and heavy episodic alcohol consumption patterns across all age groups. Furthermore, compared to obese and overweight individuals, normal and underweight individuals showed wide gaps in the mean risk of multimorbidity in the older adult age groups. There was almost a gap of five years between obese and normal persons to compare the mean risk of multimorbidity. Further, the gradient of risk in multimorbidity was maintained in obese and normal persons for the remaining years of life. Hence, in later years of life, the differentials in the risk of multimorbidity were wide in the oldest of old age groups. Furthermore, the higher risk of multimorbidity in obese persons than in normal persons was maintained throughout the life course and the gradient of risk of multimorbidity was the steepest in obese compared to other categories of BMI. Normal

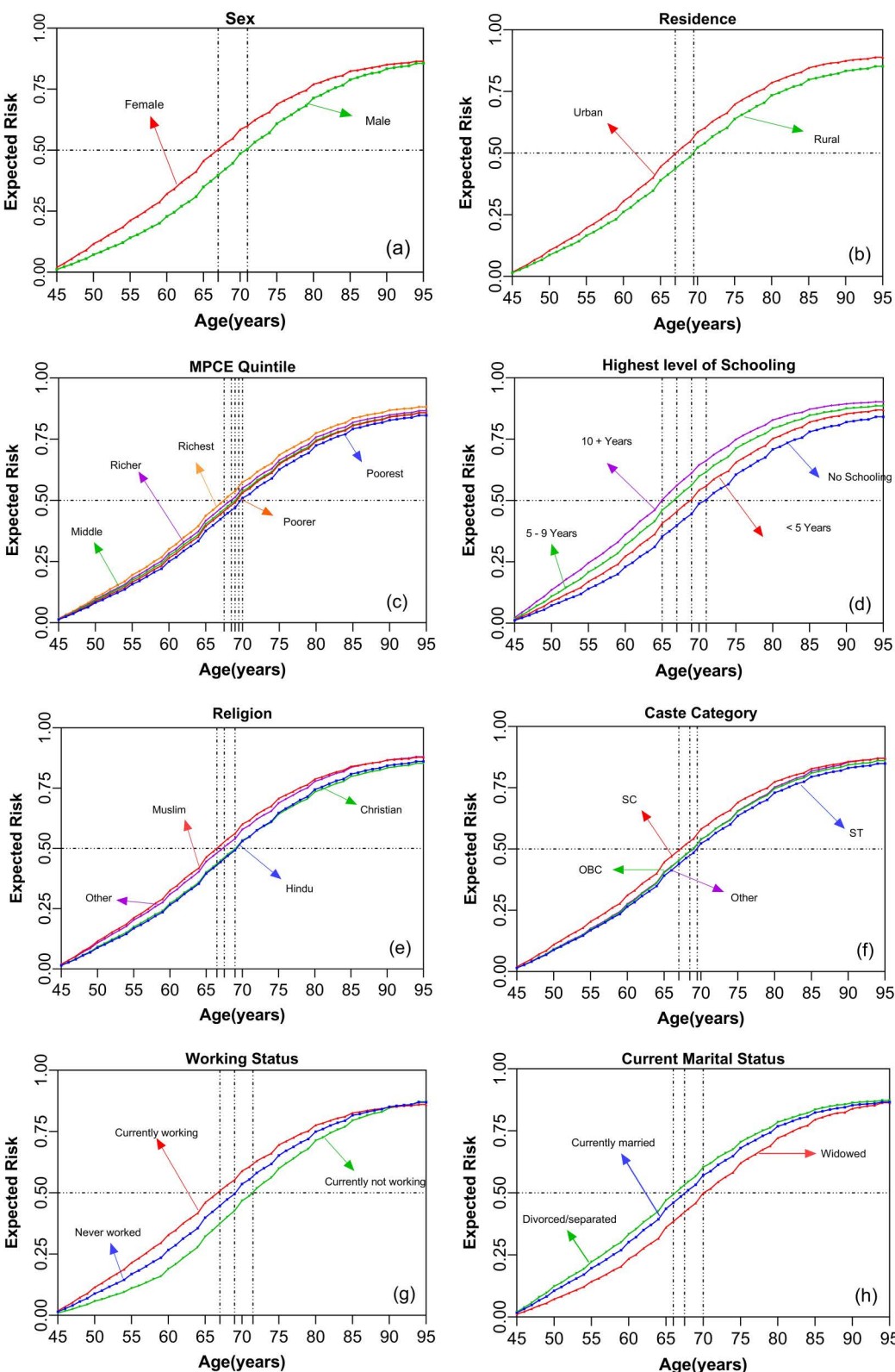

**Fig 3. Age-specific expected risk of multimorbidity susceptibility across socio-economic and demographic subgroups among adults aged 45 years and older, LASI Wave-1, 2017–18, India.**

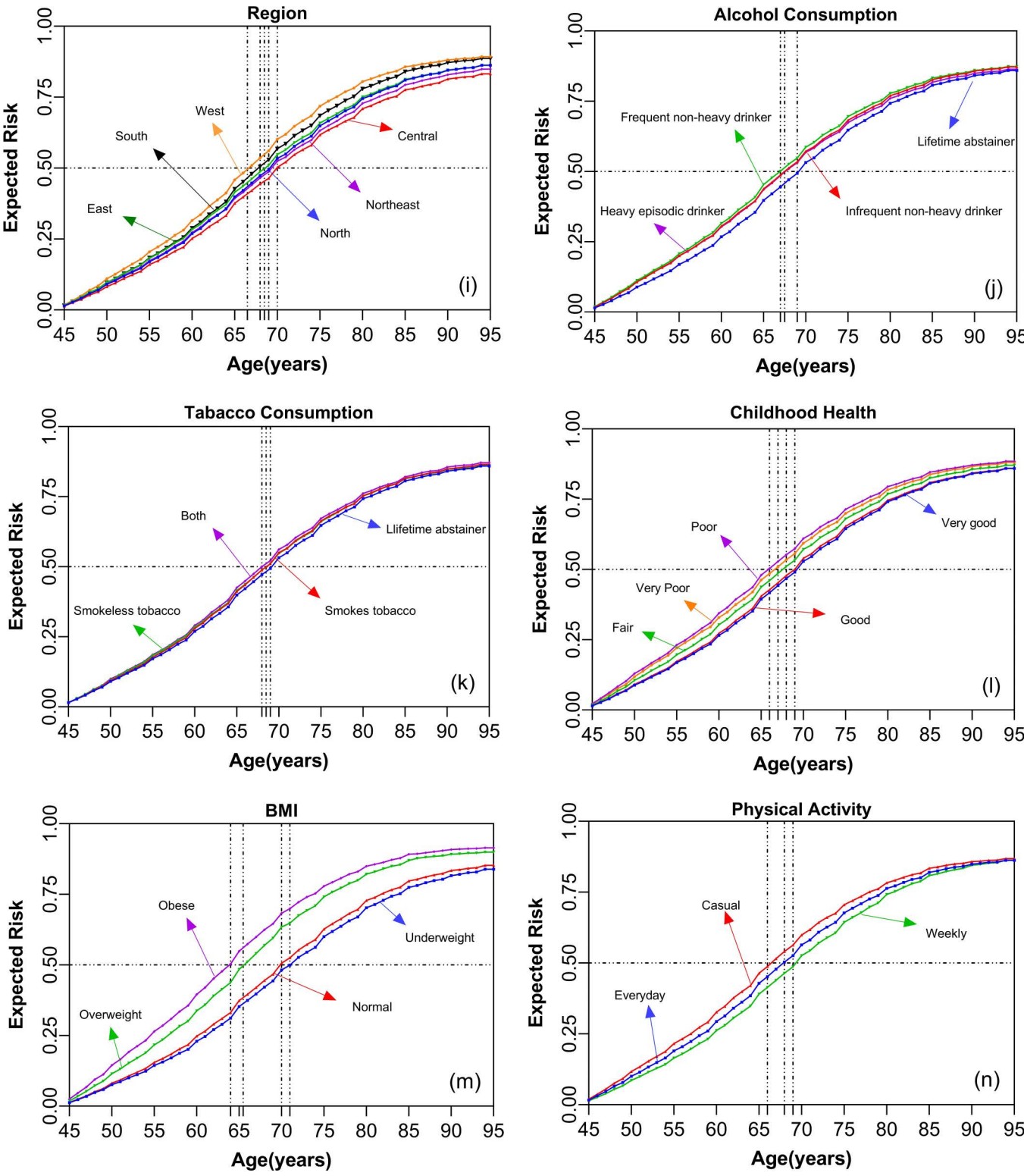

**Fig 3.** Continued.

and underweight individuals showed advantages for low risk of multimorbidity, especially in old age groups. Particularly, the differentials in the risk of multimorbidity were the widest for the BMI categories compared to other characteristics. Moreover, the RF model showed good model performance with an accuracy of 82.5%, misclassification rate of 17.5%, and an OOB error rate of 18.2%. The Cohen's Kappa was 0.74, and the AUC for multimorbidity classification was 0.89, indicating strong discriminatory power.

### 3.3 Risk of multimorbidity by leading factors

Minimal depth (VIMP or variable importance) that quantifies the total residual sum of square (RSS) decreased due to splits of given exogenous factor from the RF model (S4 Table) revealed the leading risk factors for multimorbidity susceptibility. BMI emerges as the most significant predictor of multimorbidity with the highest positive VIMP (0.06), reflecting its strong association with multimorbidity. Sex of the individuals was the second most important variable (0.05), highlights the role of gender disparities in health outcomes, which may stem from biological differences and varying access to healthcare and other lifestyle factors. Education level (0.02) also played a notable role, emphasizing its influence on health literacy, lifestyle choices, and access to preventive and curative healthcare services.

To further understand the role of multimorbidity counts, Fig 4 represents the distribution of BMI (kg/m$^2$) acrosss age, conditioned on multimorbidity counts and stratified by sex. We observed a substantial increase of multimorbidity counts (up to 7) among overweight and obese male population aged between 55–85 years. In contrast, the trend diverged to a normal and overweight population after 80 years of age. This suggests a transition of risk and augmented concentration of multimorbidity counts among the normal BMI population in the oldest of old age groups.

Nonetheless, among the female population, the prevalence of obesity was significantly higher than in males, indicating early morbidity exposure and the onset of elevated multimorbidity counts. Obesity among females resulted in increased multimorbidity counts ranging from 4 to 9 between the ages of 45 and 85 years, with the highest concentration observed in 70–75 years of age group. However, this pattern shifted in overweight and normal BMI population after 75 years of age. Among both male and female populations, the impact of obesity and overweight started to diminish after 75–80 years of age, and higher multimorbidity counts shifted to lower BMI.

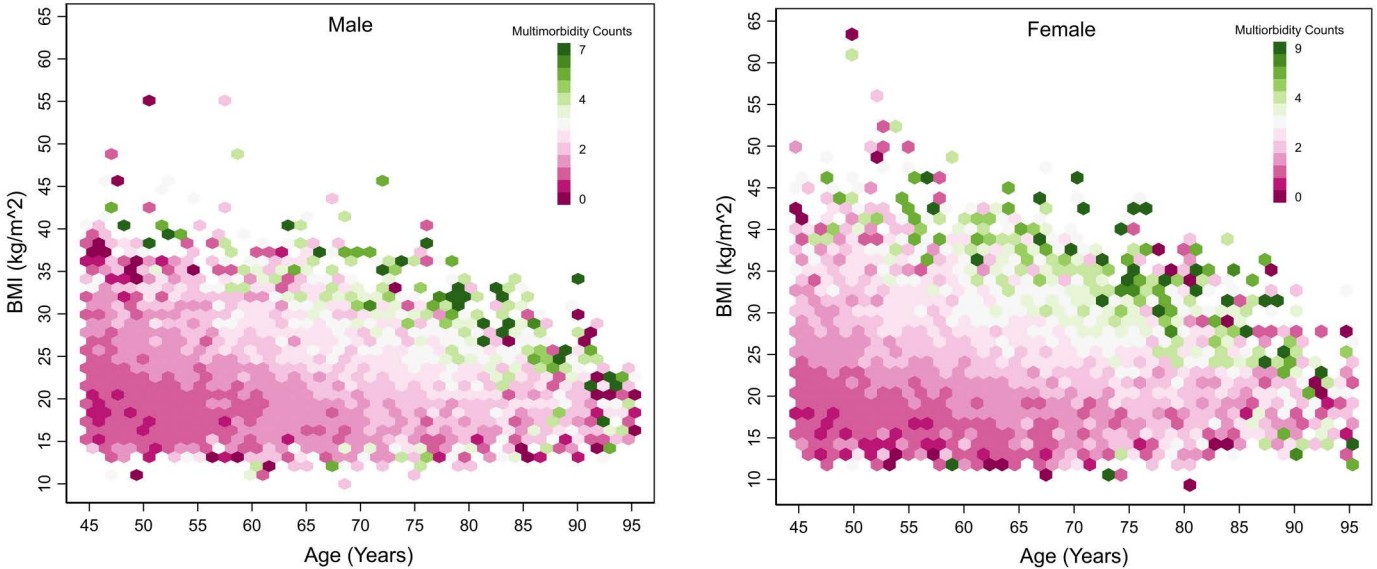

**Fig 4. Distribution of BMI (kg/m$^2$) by age, conditioned on multimorbidity counts and stratified by sex, LASI Wave-1, 2017–18, India.**

## 4 Discussion

The findings of this study highlighted crucial aspects often overlooked in public health discourse in India. First, the relative proportional share of morbidities and the compositions of multimorbidity counts over age. Second, the estimation of risk for multimorbidity susceptibility associated with socio-economic and demographic factors over age and finally, the conditional distribution of leading factors on multimorbidity counts.

Eye disorders, including cataracts, glaucoma, age-related macular degeneration (AMD), and refractive errors, are highly prevalent among older adults in high income countries [39], they are often excluded from studies of multimorbidity as a public health issue in LMICs, as they are neither degenerative nor pose a significant risk to overall health [39–41]. However, our study included eye disorders and revealed that they constitute the highest proportion of morbidity burden across all age groups, followed by CVDs and gastrointestinal conditions. According to the World Health Organization (WHO), over 2.2 billion people globally experience vision impairment or blindness, with at least one billion cases being preventable or yet to be addressed [40]. This high prevalence is particularly significant for the ageing population in India, where the risk of developing eye disorders increases with age. Despite their widespread occurrence, eye disorders were often less prioritised in health studies because they were acute conditions and perceived as less threatening to life compared to other NCDs [39,41]. However, this perspective fails to account for the substantial impact of vision impairment on an individual's quality of life [39].

Multimorbidity can be classified into two groups: complex (co-occurrence of three or more chronic conditions/diseases across >2 body systems) and non-complex (coexistence of two or more conditions/diseases) [42]. Our study demonstrated a disproportionally high complex multimorbidity burden in the older age groups. One-fourth of the population between the age of 45–49 years suffers from complex multimorbidity. The burden of complex multimorbidity rises drastically, affecting fifty percent of the population aged 85 years and above. Furthermore, complex multimorbidity is associated with a significantly higher risk of mortality, doubling the risk compared to those who do not have such a high burden of chronic conditions. Numerous studies have demonstrated that as the number of morbidities increases, so does the risk of mortality and disability [25,43,44]. Menottie (2001), specifically found that among males, complex multimorbidity increased the 10-year mortality risk across all cohorts studied [43]. Additionally, Bleijenberg et al. (2017) revealed that individuals with three or more chronic conditions were three to five times more likely to develop disabilities and functional limitations [45]. Beyond mortality and disability, complex multimorbidity also has substantial impacts on psychological and behavioural aspects such as increased levels of stress, depression, and reduced quality of life [46].

Contrary to major prior studies, which only considered the multimorbidity dyads and triads prevalence and explored only the linearly associated risk factors [47,48], this study used the novel RF model as a robust approach to analysing the gradient of multimorbidity risk across various socio-economic and demographic risk factors. RF is particularly advantageous due to its non-parametric nature, allowing it to model complex interactions between exogenous factors without specific underlying distributional assumptions. Findings from the RF model revealed that gender disparity in multimorbidity risk throughout the age of the individuals is evident, with females consistently exhibiting a higher risk in all age groups. This significant disparity also aligned with existing literature that suggested that women were more likely to report multiple chronic conditions due to longer life expectancy and greater health service utilisation [49,50]. Hormonal differences and the higher prevalence of certain diseases, such as osteoporosis and autoimmune disorders among women, further contribute to this increased burden [51,52].

BMI emerged as the second vital determinant of multimorbidity risk in our study. Underweight and normal BMI individuals had the lowest mean expected risk, while obese individuals had the highest. A similar study conducted in China showed that normal BMI reduced the risk of multimorbidity by nearly half (48%) compared to being overweight [53]. Another study revealed that overweight and obesity are associated with an increased risk of cardiometabolic multimorbidity. The risk increases from double to more than ten times in overweight and obese individuals compared to those with a normal BMI [54].

Moreover, this study further indicated that individuals with ten or more years of schooling have the highest mean expected risk of multimorbidity. Similarly, the wealth status of the individual showed the same phenomenon. Higher wealth status was associated with an increased risk of multimorbidity. These findings may seem counterintuitive but can be explained by lifestyle factors associated with higher educational levels, such as sedentary occupations, higher stress levels, and unhealthy dietary habits prevalent in professional settings [55,56]. Regional variations in multimorbidity risk across India were also significant, with the western region exhibiting the highest mean expected risk (0.54) and the Central region the lowest (0.47), can be attributed to differences in healthcare access, lifestyle factors, environmental conditions, and socio-economic development across regions [18,33]. For instance, urbanization-related lifestyle changes, pollution, and dietary shifts may contribute to higher risks in certain regions [3,4,16]. The other important finding suggested that respondent childhood health (up to age 16 years) was a significant indicator of early origins of morbidity and it has a strong association with risk of multimorbidity in the later phase of adulthood. Persons who had poor childhood health status show a high risk of multimorbidity and have maintained the gap (differences) in older-adult and old age groups in multimorbidity. Thus, morbidity, disability, and deformation at early age play vital roles in the development of multimorbidity in late adulthood.

Our findings also add depth to the understanding of the interplay between BMI and the Sex of the individuals as the leading risk factors for the distribution of complex multimorbidity. Females who were overweight or obese at a young age were more prone to a higher risk of multimorbidity than males. This disparity can be attributed to several interrelated factors including health care utilization, biological, lifestyle and psychological aspects. Hormonal changes, particularly during menopause, increase women's susceptibility to chronic conditions like cardiovascular disease and osteoporosis [57,58]. Socially, women often bear caregiving responsibilities, leading to chronic stress, which exacerbates health issues. Additionally, lifestyle factors, including sedentary behaviour and dietary habits, further contribute to their higher multimorbidity risk [59]. psychological aspects, such as higher stress levels and mental health challenges, also play a significant role. Moreover, women tend to utilise healthcare services more frequently, leading to higher reported rates of chronic conditions. Studies support our findings, showing that early-onset of obesity significantly increases the risk of complex multimorbidity in women, highlighting the need for targeted interventions to address these gender-specific health challenges [55,56,59]. Furthermore, our study shows that at later ages (80 and above), the effect of being overweight and obese gets mitigated for both genders. A study done by Tsur & Twig (2022) found that respondents aged 75 years who were struggling with obesity had an estimated incidence of complex multimorbidity of 8.3%, while among those with a healthy weight, it was only 1.0%. Further, Obesity was associated with a 12.4 times higher risk of complex multimorbidity in the Finnish cohorts, with a population-attributable fraction of 55.2% [60,61]. The risk of complex multimorbidity was greater in participants who had obesity at younger than 50 years compared to those who developed obesity later in life [60]. Another study demonstrated that women with early-onset obesity were more likely to develop multiple chronic conditions, including cardiovascular diseases, diabetes, and musculoskeletal disorders, compared to their male counterparts [62]. Similarly, Blumel et al. (2017) found that midlife obesity was strongly associated with an increased risk of complex multimorbidity in women, emphasising the need for targeted interventions [63]. Moreover, a recent evidence further underscores the role of multimorbidity not only in diminishing quality of life among older women but also in mediating the impact of exogenous socioeconomic and demographic determinants [64].This highlights a transition of the risk of complex multimorbidity from obese to normal BMI individuals in the oldest of old age groups, indicating the critical importance of addressing sex-specific and age-related factors in understanding and mitigating multimorbidity risk.

Despite the novelty of our analysis and approach, which integrated evidence from robust and new statistical methods with a unique methodology, our study had few limitations. This study included a more comprehensive list of morbidities that has not been considered in any prior studies of multimorbidity in India, yet it was not possible to include functional limitation, Activities of daily living (ADL) and Instrumental activities of daily living (IADL) as these have a confounding relationship with multimorbidity. The marked heterogeneity in multimorbidity estimates and frequent reliance on self-reported conditions makes it challenging to draw comparisons between studies. Moreover, 45 years and above adults were considered in the

study, which may have reduced variability and could potentially increase the difficulty of identifying an association between factors and multimorbidity. To address the issue of overadjustment, the risk estimates of multimorbidity are adjusted only for age, as it exhibits a strong correlation with both exogenous factors and multimorbidity. Another limitation of our study was that while we categorize diseases into degenerative, non-communicable, chronic, and communicable, our measure of complex multimorbidity does not allow us to distinguish how many of these conditions are chronic within a given combination. This limits our ability to assess the specific burden of chronic diseases within complex multimorbidity patterns.

The prevalence of multimorbidity in older persons ranges from 55 to 98%. In cross-sectional studies, older age, female gender, and low socio-economic status are factors associated with multimorbidity, confirmed by longitudinal studies as well. The major consequences of multimorbidity are disability, functional decline, poor quality of life, and high healthcare costs. Methodological issues in evaluating multimorbidity are discussed, as well as future research needs, especially concerning etiological factors, combinations and clustering of chronic diseases, and care models for persons affected by multiple disorders. New insights in this field can lead to the identification of preventive strategies and better treatment of patients with multimorbidity.

## 5 Conclusion

The results of this study illustrate a comprehensive analysis of morbidity and multimorbidity among older adults in India. Eye disorders and CVDs were the leading contributors to the wide spectrum of disease burden across all age groups, followed by endocrine diseases, gastrointestinal conditions, and infectious diseases, with sequentially converging burden in later years. While severe diseases such as cancer and chronic lung diseases maintained a constant presence. Multimorbidity analysis revealed a higher prevalence of two to four conditions across age groups, with a significant increase in five or more conditions, peaking in the 70–74 age group.

Notably, women exhibited a higher risk of multimorbidity compared to men, with the risk gap widening with age. This study also highlighted the influence of socio-economic and demographic factors on multimorbidity. For instance, individuals with higher levels of education, obesity, and poor childhood health have a higher risk of multimorbidity at earlier ages. Physical activity demonstrates a protective effect, reducing the risk of multimorbidity, while casual or no physical activity increases the risk.

Body Mass Index (BMI) has emerged as the leading significant factor, such that overweight and obese individuals are at a higher risk of multimorbidity susceptibility compared to their normal and underweight counterparts. The differential in multimorbidity risk was stark, with a noticeable gap between obese and normal-weight individuals.

Overall, this study underscores the complex interplay between ageing, socio-economic factors, and lifestyle factors in shaping the burden of multimorbidity among older adults in India, calling for targeted public health interventions to address these disparities and further promote healthier ageing.

## Supporting information

**S1 Table. Morbidity and multimorbidity prevalence among older adults, Longitudinal Ageing Study in India (LASI), wave-1, 2017–2018.**
(DOCX)

**S2 Table. Detailed descriptions of all the exogenous factors, Longitudinal Ageing Study in India (LASI), wave-1, 2017–2018.**
(DOCX)

**S3 Table. Multimorbidity prevalence by exogenous factors among older adults, Longitudinal Ageing Study in India (LASI), wave-1, 2017–2018.**
(DOCX)

**S4 Table. VIMP from RF analysis of multimorbidity and no multimorbidity, Longitudinal Ageing Study in India (LASI), wave-1, 2017–2018.**
(DOCX)

## Acknowledgments

Not applicable.

## Author contributions

**Conceptualization:** Ajay Kumar, Bharti Singh.

**Data curation:** Ajay Kumar.

**Formal analysis:** Ajay Kumar.

**Methodology:** Ajay Kumar, Bharti Singh.

**Resources:** Bharti Singh.

**Supervision:** Ajay Kumar, Bharti Singh.

**Validation:** Ajay Kumar, Bharti Singh.

**Visualization:** Ajay Kumar.

**Writing – original draft:** Ajay Kumar, Bharti Singh.

**Writing – review & editing:** Ajay Kumar, Bharti Singh.

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
