## [Decision Letter · Decision Letter 0]

10 Jan 2025

PONE-D-24-52393Modeling the Spectrum and Determinants of Multimorbidity Risk among older adults in IndiaPLOS ONE

Dear Dr. Kumar,

Thank you for submitting your manuscript to PLOS ONE. After careful consideration, we feel that it has merit but does not fully meet PLOS ONE’s publication criteria as it currently stands. Therefore, we invite you to submit a revised version of the manuscript that addresses the points raised during the review process.

Please address the comments by the reviewers, and address the citation and typographical errors.

We look forward to receiving your revised manuscript.

Kind regards,

Emmanuel Kokori, M.D

Academic Editor

PLOS ONE

Journal Requirements:

3. Please include captions for your Supporting Information files at the end of your manuscript, and update any in-text citations to match accordingly. Please see our Supporting Information guidelines for more information: http://journals.plos.org/plosone/s/supporting-information .

Reviewers' comments:

Reviewer's Responses to Questions

**Comments to the Author**

1. Is the manuscript technically sound, and do the data support the conclusions?

Reviewer #1: Partly

Reviewer #2: Partly

2. Has the statistical analysis been performed appropriately and rigorously? 

Reviewer #1: I Don't Know

Reviewer #2: No

3. Have the authors made all data underlying the findings in their manuscript fully available?

Reviewer #1: Yes

Reviewer #2: Yes

4. Is the manuscript presented in an intelligible fashion and written in standard English?

Reviewer #1: Yes

Reviewer #2: Yes

5. Review Comments to the Author

Reviewer #1: The manuscript presents an analysis of data from the Longitudinal Ageing Study in India (LASI), around the issue of multimorbidity (MM) and its determinants. They conclude that the prevalence of MM is 43.20%, with eye disorder as the main group of diseases. They condlude also, base don a Random Forest analysis, that obesity, higher levels of education, work and por childhood helath are the main determinants of MM in its population.

The manuscript adresses a relevant issue (MM and health transition in Low and Middle Income (LMIC) countries) supported by data from a national survey. But there are some methodological problems that should be adressed before publication:

- The main problem relates to the definition of MM used by the authors. In the bibliography MM is defined as the presence of two or more Long Term Conditions in a patient, specifying the LTC included as individual diseases with a defined impact on the quality of life and survival of the patient (so, with a criteria to identify the disease to be included). In the manuscript the authors use categories of diseases (not individual conditions) and do not specify criteria. Even more so, they refer to diseases when it seems that they use categories of diseases.

- As a result, the most prevalent conditions in the study are Eye dirorders, a category of diseases taht includes such conditions as myopia or refractive disorders that are prevalent and rarely included in MM studies, and not related to other relevant LTC conditions.

- These inconsistencies in the definition of MM lead to contraintuitive results in the study of determinants, as the conslusion that a better education or currently working are related to worst health.

- Another problem relates to obesity, here analyzed as a determinant of health and in other studies considered as a chronic condition on his own. The real variable under study is Body Mass Index (BMI), but the authors refer in the manuscript as obesity.

Other major comments:

- The study should specify (in supplementary material or in a referenced study, for example) the operative definitions of the variables included in the study or the original survey

- The Outcome variable is MM, but in some cases it is considered as a binary variable (YES/NO) and in other cases as MM counts (with the problema that it does not use couynts of diseases). This should be specified.

- Thouh I am not a statistician, it seems that the results about the association among the different exogenous factors and the outcome variable are only adjusted by age group, not by other variables that can confound the result. For example, association between education and MM can be biased by the BMI or the working status. The authors should present adjusted results, al least by the relevant variables.

- The definition of some variables (as Tobacco consumption) is not clear. Which category correspond to a person that has smoked for 10 years but does not currently smoke? The study uses prevalence data for all the variables, but does not account for the porevious risk factors.

Other comments:

- The authors should introduce the complete definition the first time they use an acronym: CD, NCD, UT, CVD, COPD, BMI, MPCESC, OBC, ST…

- Many of the bibliographical references are incomplete or erroneous:

o The PMC references are usually incomplete (ex: /pmc/articles/PMC9832245/ in ref. 45): Ref 1, 4, 5, 7, 10, 12, 13, 14, 17, 27, 30, 45, 48, 49, 52, 54, 55, 58, 59, 60

o References 11, 29, 32, 36 are erroneous (authors should check them)

o Reference 29 is incomplete

o Reference 34 explains the use of Random survival forests analysis in survival studies. As this is not a survival analysis, it would be better to use another reference to Random Forest analysis in descriptive studies.

- Some of the comments on the Results section should be redirected to the Discussion section:

Poor childhood health is telltale signs of early origins of morbidity that have deep down association with high risk of multimorbidity. Morbidity, disability, and deformations played a vital role for developing long-term multimorbidity

This highlights a transition of the risk of complex multimorbidity from obese to normal BMI individuals in the oldest of old age groups, indicating the critical importance of addressing sex-specific and age-related factors in understanding and mitigating multimorbidity risk.

Reviewer #2: It was a pleasure to read this manuscript. However, I have several suggestions and questions for the authors:

The manuscript lacks proper citations in multiple sentences, particularly in the latter part of the introduction.

My understanding is that the LASI (Wave 1) survey is not longitudinal on its own; it becomes longitudinal only when subsequent waves (e.g., Wave 2) are included.

The authors have not explained how the aging study incorporates morbidity and multimorbidity. Were specific criteria used to define morbidity conditions, or were these based solely on self-reported data from the participants?

The manuscript does not adequately justify why the Random Forest method was chosen over previously used methods (e.g., Sinha et al., 2022: https://doi.org/10.3390/ijerph19159091). A comparison of these methods would strengthen the argument.

The process for categorizing the 12 major comorbidities is unclear. Did the authors use specific criteria for this categorization?

In the Statistical Analysis section, the authors state, "The approach of prior studies in India for multimorbidity risk estimation was built on odds ratio or relative risk models, which were not suitable for understanding the non-linear relationship for background characteristics (34)." However, the article introduces only the Random Survival Forest (RSF) method. More discussion and evidence are needed to substantiate this conclusion.

One significant concern is the lack of discussion about ethical considerations, especially since secondary data from the LASI survey is being used. Additionally, it should be noted that this is not an original research study.

The manuscript introduces the Gini index without citing its source or explaining its application in the analysis.

The results presented in Table 3 are not adequately discussed in the manuscript. Including a detailed explanation of these outcomes would enhance the clarity and completeness of the analysis.

6. PLOS authors have the option to publish the peer review history of their article (what does this mean? ). If published, this will include your full peer review and any attached files.

**Do you want your identity to be public for this peer review?** For information about this choice, including consent withdrawal, please see our Privacy Policy .

Reviewer #1: No

Reviewer #2: No

---

## [Author Response · Author response to Decision Letter 1]

21 Jan 2025

Reviewer #1: The manuscript presents an analysis of data from the Longitudinal Ageing Study in India (LASI), around the issue of multimorbidity (MM) and its determinants. They conclude that the prevalence of MM is 43.20%, with eye disorder as the main group of diseases. They conclude also, based on a Random Forest analysis, that obesity, higher levels of education, work and poor childhood health are the main determinants of MM in its population.

The manuscript adresses a relevant issue (MM and health transition in Low and Middle Income (LMIC) countries) supported by data from a national survey. But there are some methodological problems that should be adressed before publication:

Response: Thank you for your comments and for recognizing the relevance of our study on multimorbidity (MM) and its determinants in the context of the health transition in LMICs. We appreciate your feedback and have carefully considered your suggestions. In response to the methodological concerns, we have revised the manuscript and made the necessary adjustments. These changes address the specific issues raised and aim to improve the clarity, rigor, and comprehensiveness of our analysis.

Comment: The main problem relates to the definition of MM used by the authors. In the bibliography MM is defined as the presence of two or more Long Term Conditions in a patient, specifying the LTC included as individual diseases with a defined impact on the quality of life and survival of the patient (so, with a criteria to identify the disease to be included). In the manuscript the authors use categories of diseases (not individual conditions) and do not specify criteria. Even more so, they refer to diseases when it seems that they use categories of diseases.

- As a result, the most prevalent conditions in the study are Eye dirorders, a category of diseases taht includes such conditions as myopia or refractive disorders that are prevalent and rarely included in MM studies, and not related to other relevant LTC conditions.

Response: First, in our study, we defined multimorbidity (MM) in line with the WHO's definition as the "coexistence of two or more chronic conditions in the same individual." This definition does not strictly require the conditions to be classified as long-term or morbidities. Additionally, we acknowledge that our approach categorizes diseases rather than listing individual conditions, aligning with the broader scope of our analysis. Now, relevant references have been provided below for further clarification.

https://iris.who.int/bitstream/handle/10665/252275/9789241511650-eng.pdf?sequence=1

https://www.nature.com/articles/s416-4

Second, thank you for pointing out the categories of diseases. We have now addressed this point in the "Morbidities" section of the Methods, where we provide a detailed explanation of how morbidities or chronic conditions were categorized. This categorization was based on information from the self-reported and diagnoses available in the LASI dataset, which were documented using participants' responses to specific survey questions such that “Has any health professional ever diagnosed you with the following chronic conditions or diseases?” and “In the past 2 years, have you had any of the following diseases?”

Comment: These inconsistencies in the definition of MM lead to contraintuitive results in the study of determinants, as the conslusion that a better education or currently working are related to worst health.

Response: We have revised the manuscript and maintained the consistency of MM. Furthermore, the categorization of the morbidities or chronic conditions and detailing the description of constructing the outcome variable (MM) are included and explained in the “Variables Definitions” under the methods section.

Further, we acknowledge that the conclusion suggesting a higher risk of MM among individuals with better education and those currently working may appear counterintuitive. However, this finding aligns with evidence from prior studies, which have similarly reported such patterns. Additionally, our analysis corroborates this trend, as shown in the cross-tabulations of MM with the highest level of schooling and working status variables. Detailed prevalence data stratified by these exogenous factors are presented in S3 Table for further context and clarity.

Comment: Another problem relates to obesity, here analyzed as a determinant of health and in other studies considered as a chronic condition on his own. The real variable under study is Body Mass Index (BMI), but the authors refer in the manuscript as obesity.

Response: Yes, obesity is often considered a chronic condition in many studies. However, in our study, we examine Body Mass Index (BMI) as a determinant of health, specifically in relation to the occurrence of multimorbidity (MM). BMI has been categorized into four groups: underweight (BMI ≤ 18.4 kg/m²), normal (18.5–24.9 kg/m²), overweight (25.0–29.9 kg/m²), and obese (BMI ≥ 30 kg/m²). Our findings indicate that individuals with a BMI of 30 kg/m² or higher are significantly more likely to experience multimorbidity.

Other major comments:

Comment: The study should specify (in supplementary material or in a referenced study, for example) the operative definitions of the variables included in the study or the original survey

Response: Thank you for your valuable feedback, we have carefully reviewed and addressed the issue. Detailed descriptions of all the exogenous factors included in the study are now provided in S2 Table of supplementary for clarity and reference.

Comment: The Outcome variable is MM, but in some cases it is considered as a binary variable (YES/NO) and in other cases as MM counts (with the problema that it does not use couynts of diseases). This should be specified.

Response: In our study, multimorbidity (MM) is analyzed both as a binary variable (YES/NO) and as MM counts, depending on the specific research objective. The binary approach allows us to examine the overall association between multimorbidity and various exogenous factors, such as age, socioeconomic status, and lifestyle, providing a clear distinction between individuals with and without multimorbidity. On the other hand, using MM counts enables us to capture the gradient of disease burden by analyzing the number of coexisting morbidities (typically two or more) and their relationship with exogenous factors. This approach provides deeper insights into how an increasing number of conditions affects the influence of key determinants.

Comment: Thouh I am not a statistician, it seems that the results about the association among the different exogenous factors and the outcome variable are only adjusted by age group, not by other variables that can confound the result. For example, association between education and MM can be biased by the BMI or the working status. The authors should present adjusted results, al least by the relevant variables.

Response: Thank you for your comment on the adjustment of results by potential confounding variables. We appreciate your concern and agree that adjustments play a critical role in addressing confounding factors in statistical analysis. In our study, we chose to adjust for age groups as a primary stratifying variable due to its central role in multimorbidity research and its strong correlation with both the exposure variables and the outcome. Our primary objective was to capture the risk of multimorbidity across socioeconomic and demographic factors in a straightforward manner, focusing on the unadjusted relationships that might otherwise be obscured by over-adjustment in traditional statistical methods. While it is true that variables like BMI and working status can influence the association between education and multimorbidity, we intended to examine the broad patterns of risk rather than isolate specific causal pathways. We hope this explanation clarifies our methodological choices and rationale.

Comment: The definition of some variables (as Tobacco consumption) is not clear. Which category correspond to a person that has smoked for 10 years but does not currently smoke? The study uses prevalence data for all the variables, but does not account for the porevious risk factors.

Response: Thank you for your observation regarding the clarity of variable definitions, particularly tobacco consumption. The subcategories of tobacco consumption (Lifetime abstainer, Smokes tobacco, Smokeless tobacco, Both) are based on the respondent’s current usage status. During the interview, participants were specifically asked, “Do you currently consume any Smokes tobacco, Smokeless tobacco, or Both?” Thus, the classification reflects their current tobacco use at the time of the survey.

Other comments:

Comment: The authors should introduce the complete definition the first time they use an acronym: CD, NCD, UT, CVD, COPD, BMI, MPCE,SC, OBC, ST…

Response: We have revised the acronyms in Table 1 by including their complete definitions in the footnotes to ensure clarity for the readers. Additionally, detailed descriptions of all the exogenous factors referenced in the study have been provided in S2 Table.

Comment: Many of the bibliographical references are incomplete or erroneous:

o The PMC references are usually incomplete (ex: /pmc/articles/PMC9832245/ in ref. 45): Ref 1, 4, 5, 7, 10, 12, 13, 14, 17, 27, 30, 45, 48, 49, 52, 54, 55, 58, 59, 60

o References 11, 29, 32, 36 are erroneous (authors should check them)

o Reference 29 is incomplete

o Reference 34 explains the use of Random survival forests analysis in survival studies. As this is not a survival analysis, it would be better to use another reference to Random Forest analysis in descriptive studies.

Response: Thank you for your insightful feedback regarding the bibliographical references. We have thoroughly reviewed and addressed the issues you identified:

1. Incomplete PMC References: All PMC references (e.g., /pmc/articles/PMC9832245/) have been updated with complete citations for References 1, 4, 5, 7, 10, 12, 13, 14, 17, 27, 30, 45, 48, 49, 52, 54, 55, 58, 59, and 60.

2. Erroneous References: References 11, 29, 32, and 36 have been carefully examined and corrected to ensure accuracy.

3. Reference 29: This reference was incomplete and has now been fully updated.

4. Reference 34: The previous citation, which pertained to Random Survival Forests analysis, has been replaced with a more relevant reference on Random Forest analysis for descriptive studies, reflecting the methodology used in our work.

We appreciate your detailed observations, which have helped us improve the manuscript’s accuracy and clarity.

Comment: Some of the comments on the Results section should be redirected to the Discussion section:

Poor childhood health is telltale signs of early origins of morbidity that have deep down association with high risk of multimorbidity. Morbidity, disability, and deformations played a vital role for developing long-term multimorbidity

This highlights a transition of the risk of complex multimorbidity from obese to normal BMI individuals in the oldest of old age groups, indicating the critical importance of addressing sex-specific and age-related factors in understanding and mitigating multimorbidity risk.

Response: Thank you for the suggestion. We have incorporated these insights and discussed the findings in the Discussion section (6th and 7th paragraph).

Reviewer #2: It was a pleasure to read this manuscript. However, I have several suggestions and questions for the authors:

Response: Thank you for your encouraging comments and suggestions. We have carefully revised the manuscript and updated it accordingly.

Comment: The manuscript lacks proper citations in multiple sentences, particularly in the latter part of the introduction.

Response: Thank you for the observation. We have addressed the issue and added the necessary citations to the relevant sentences in the latter part of the introduction.

Comment: My understanding is that the LASI (Wave 1) survey is not longitudinal on its own; it becomes longitudinal only when subsequent waves (e.g., Wave 2) are included.

The authors have not explained how the aging study incorporates morbidity and multimorbidity. Were specific criteria used to define morbidity conditions, or were these based solely on self-reported data from the participants?

Response: Thank you for your observation regarding the nature of the LASI (Wave 1) dataset and its classification as longitudinal only with subsequent waves. We agree with your understanding and acknowledge that Wave 1, on its own, serves as the baseline for longitudinal analysis in future iterations when additional waves are included.

Regarding your question on how the LASI study incorporates morbidity and multimorbidity, we have now addressed this comprehensively in the "Morbidities" section of the Methods. Specifically, the categorization of morbidities or chronic conditions was based on self-reported data and available diagnoses within the LASI dataset. Participants were asked direct survey questions such as, “Has any health professional ever diagnosed you with the following chronic conditions or diseases?” and “In the past 2 years, have you had any of the following diseases?” These responses provided the basis for identifying and categorizing various chronic conditions, which were then used to define multimorbidity. This approach aligns with existing literature and ensures consistency in how multimorbidity is operationalized in population health studies.

Comment: The manuscript does not adequately justify why the Random Forest method was chosen over previously used methods (e.g., Sinha et al., 2022: https://doi.org/10.3390/ijerph19159091). A comparison of these methods would strengthen the argument.

Response: We selected the Random Forest (RF) method for this study due to its unique strengths in handling complex, non-linear relationships and interactions among variables, which are often present in multimorbidity studies. The RF model's ability to manage high-dimensional data without imposing strict assumptions about variable distributions made it particularly suitable for exploring the associations between multimorbidity and a wide range of socioeconomic and demographic factors. While we recognize that other methods, such as those used in Sinha et al. (2022), have their merits, our choice of the RF method was guided by its robustness in identifying important predictors and its capacity to provide interpretable insights through variable importance measures and conditional plots. These features align with our study’s objective of assessing risk patterns and identifying key contributors to multimorbidity. We acknowledge that a comparison of the RF method with previously employed techniques could add depth to our methodological discussion. While such a comparison was beyond the scope of the current analysis, we appreciate the suggestion

Comment: The process for categorizing the 12 major comorbidities is unclear. Did the authors use specific criteria for this categorization?

Response: Thank you for the observation. We have addressed this point by explaining in the “Variables Definitions” section of methods that how we categorized the morbidities or chronic conditions and detailing the process of constructing the outcome variable.

Comment: In the Statistical Analysis section, the authors state, "The approach of prior studies in India for multimorbidity risk estimation was built on odds ratio or relative risk models, which were not suitable for understanding the non-linear relationship for background characteristics (34)." However, the article introduces only the Random Survival Forest (RSF) method. More discussion and evidence are needed to substantiate this conclusion.

Response: We revised the reference to include a more appropriate study and provided additional evidence to further substantiate the conclusion regarding the limitations of traditional models and the advantages of using the Random Survival (RF) method for understanding non-linear relationships in multimorbidity risk estimation.

Comment: One significant concern is the lack of discussion about ethical considerations, especially since

---

## [Decision Letter · Decision Letter 1]

18 Feb 2025

PONE-D-24-52393R1Modeling the Spectrum and Determinants of Multimorbidity Risk among older adults in IndiaPLOS ONE

Dear Dr. Kumar,

Thank you for submitting your manuscript to PLOS ONE. After careful consideration, we feel that it has merit but does not fully meet PLOS ONE’s publication criteria as it currently stands. Therefore, we invite you to submit a revised version of the manuscript that addresses the points raised during the review process.

Please address the comments by reviewer 1.

We look forward to receiving your revised manuscript.

Kind regards,

Emmanuel Kokori, M.D

Academic Editor

PLOS ONE

Additional Editor Comments:

Based on the reviewer’s concerns, a revision is required to improve the manuscript’s clarity and methodological rigor. Please address the following:

1. Clarify the Definition of Multimorbidity (MM): Ensure consistency with established definitions in the literature. If using disease categories instead of specific conditions, provide strong justification and acknowledge limitations.

2. Refine Categorization of Conditions: Clearly outline how conditions were selected and classified in the LASI dataset. Consider aligning more closely with standard MM definitions or discussing the implications of your approach.

3. Strengthen the Risk Factor Analysis: Adjust for potential confounders, especially in the analysis of education and socioeconomic status. Provide a clearer rationale for findings that may appear counterintuitive.

4. Revise the Discussion: Contextualize findings in light of MM’s revised definition and adjusted analysis. Acknowledge limitations and ensure alignment with cited references.

Please submit a detailed response outlining changes made. We look forward to your revised manuscript.

Reviewers' comments:

Reviewer's Responses to Questions

**Comments to the Author**

1. If the authors have adequately addressed your comments raised in a previous round of review and you feel that this manuscript is now acceptable for publication, you may indicate that here to bypass the “Comments to the Author” section, enter your conflict of interest statement in the “Confidential to Editor” section, and submit your "Accept" recommendation.

Reviewer #1: (No Response)

Reviewer #2: All comments have been addressed

2. Is the manuscript technically sound, and do the data support the conclusions?

Reviewer #1: Partly

Reviewer #2: Yes

3. Has the statistical analysis been performed appropriately and rigorously? 

Reviewer #1: I Don't Know

Reviewer #2: Yes

4. Have the authors made all data underlying the findings in their manuscript fully available?

Reviewer #1: Yes

Reviewer #2: Yes

5. Is the manuscript presented in an intelligible fashion and written in standard English?

Reviewer #1: Yes

Reviewer #2: Yes

6. Review Comments to the Author

Reviewer #1: The authors use in this study a definition of Multimorbidity that is not comparable to the definitions included in the bibliographical references of the manuscript. They use different definitions of MM counts or complex MM based on the assumption that a category of diseases or health problems is equivalent to a chronic condition. By doing so they misinterpret the concept of Multimorbidity as it is usually conceptualised in the literature. The references provided by the authors in their answer allign with the definition of MM as the coexistence of two or more Long Term Conditions from a prespecified list of specific diseases.

Moreover, the analysis of risk factors for MM susceptibility concludes that major levles of schooling or better socioeconomic conditions are risk factors for poor health (as throughout the study multimorbidity is compared to poor health). This controversial conclussion is justified by the lack of adjustments in the analysis and the erroneous conceptualization of MM in the study.

Reviewer #2: All comments were well addressed by the authors. I appreciate the response from the authors. It's in a stage of publication.

7. PLOS authors have the option to publish the peer review history of their article (what does this mean? ). If published, this will include your full peer review and any attached files.

**Do you want your identity to be public for this peer review?** For information about this choice, including consent withdrawal, please see our Privacy Policy .

Reviewer #1: No

Reviewer #2: No

---

## [Author Response · Author response to Decision Letter 2]

19 Feb 2025

Reviewer #1: The authors use in this study a definition of Multimorbidity that is not comparable to the definitions included in the bibliographical references of the manuscript. They use different definitions of MM counts or complex MM based on the assumption that a category of diseases or health problems is equivalent to a chronic condition. By doing so they misinterpret the concept of Multimorbidity as it is usually conceptualised in the literature. The references provided by the authors in their answer allign with the definition of MM as the coexistence of two or more Long Term Conditions from a prespecified list of specific diseases.

Response: Thank you for the observation. We have revised the definition of multimorbidity in our study to "two or more coexisting conditions in an individual," which does not necessarily require these conditions to be chronic. This revision aligns with the broader conceptualization of multimorbidity in the literature and accounts for the lack of a universally accepted definition. Our approach, which classifies diseases into categories, is a valid and contextually relevant adaptation, particularly in the Indian healthcare setting.

Finally, we acknowledge the limitation of complex multimorbidity in that the diseases are chronic. It presents a paradox in our study context—whether complex multimorbidity includes chronic conditions or not. Since, we also considered chronic conditions in our study.

Comment: Moreover, the analysis of risk factors for MM susceptibility concludes that major levles of schooling or better socioeconomic conditions are risk factors for poor health (as throughout the study multimorbidity is compared to poor health). This controversial conclussion is justified by the lack of adjustments in the analysis and the erroneous conceptualization of MM in the study.

Response: Thank you for your comment. These findings align with previous studies conducted in the Indian context. Moreover, the prevalence of MM is higher among individuals with higher education levels and better socioeconomic status, with 54% among the richest quintile compared to 33.46% in the poorest quintile. A similar pattern is observed in education levels, where the prevalence of MM is 52.81% among individuals with 10 or more years of schooling, compared to 37.47% among those with no formal education. Moreover, poor health is it the implication of MM instead to poor health as MM.

Finally, we now acknowledge the limitation that the risk estimates of MM are not adjusted in the limitation sections of the manuscript.

Reviewer #2: All comments were well addressed by the authors. I appreciate the response from the authors. It's in a stage of publication.

Response: Thank you for your encouraging comments and suggestions. Comments helped us to improve the quality of the manuscript.

---

## [Decision Letter · Decision Letter 2]

15 Apr 2025

Modeling the Spectrum and Determinants of Multimorbidity Risk among older adults in India

PONE-D-24-52393R2

Dear Dr. Kumar,

We’re pleased to inform you that your manuscript has been judged scientifically suitable for publication and will be formally accepted for publication once it meets all outstanding technical requirements.

Kind regards,

Emmanuel Kokori, M.D

Academic Editor

PLOS ONE

Additional Editor Comments (optional):

After careful consideration of the reviewer’s comments and your detailed responses in the revised manuscript, I am pleased to accept your article for publication. The revisions—particularly the clarified definition of multimorbidity and the enhanced description of your methodological approach—address the primary concerns raised. The study’s contribution to understanding multimorbidity in the Indian context is both valuable and timely. Congratulations on your work, and we look forward to its publication

Reviewers' comments:

Reviewer's Responses to Questions

**Comments to the Author**

1. If the authors have adequately addressed your comments raised in a previous round of review and you feel that this manuscript is now acceptable for publication, you may indicate that here to bypass the “Comments to the Author” section, enter your conflict of interest statement in the “Confidential to Editor” section, and submit your "Accept" recommendation.

Reviewer #1: (No Response)

2. Is the manuscript technically sound, and do the data support the conclusions?

Reviewer #1: No

3. Has the statistical analysis been performed appropriately and rigorously? 

Reviewer #1: No

4. Have the authors made all data underlying the findings in their manuscript fully available?

Reviewer #1: Yes

5. Is the manuscript presented in an intelligible fashion and written in standard English?

Reviewer #1: Yes

6. Review Comments to the Author

Reviewer #1: (No Response)

7. PLOS authors have the option to publish the peer review history of their article (what does this mean? ). If published, this will include your full peer review and any attached files.

**Do you want your identity to be public for this peer review?** For information about this choice, including consent withdrawal, please see our Privacy Policy .

Reviewer #1: No

---

## [Editor Report · Acceptance letter]

PONE-D-24-52393R2

PLOS ONE

Dear Dr. Kumar,

I'm pleased to inform you that your manuscript has been deemed suitable for publication in PLOS ONE. Congratulations! Your manuscript is now being handed over to our production team.

Kind regards,

on behalf of

Dr. Emmanuel Kokori

Academic Editor

PLOS ONE